# Low-Pass Genome Sequencing-Based Detection of Paternity: Validation in Clinical Cytogenetics

**DOI:** 10.3390/genes14071357

**Published:** 2023-06-27

**Authors:** Keying Li, Yilin Zhao, Matthew Hoi Kin Chau, Ye Cao, Tak Yeung Leung, Yvonne K. Kwok, Kwong Wai Choy, Zirui Dong

**Affiliations:** 1Department of Obstetrics and Gynaecology, The Chinese University of Hong Kong, Hong Kong, China; keyingli@cuhk.edu.hk (K.L.); zhaoyilin@link.cuhk.edu.hk (Y.Z.); matthewchau@cuhk.edu.hk (M.H.K.C.); yecao@cuhk.edu.hk (Y.C.); tyleung@cuhk.edu.hk (T.Y.L.); kky254@ha.org.hk (Y.K.K.); richardchoy@cuhk.edu.hk (K.W.C.); 2Shenzhen Research Institute, The Chinese University of Hong Kong, Shenzhen 518057, China; 3Hong Kong Hub of Paediatric Excellence, The Chinese University of Hong Kong, Hong Kong, China; 4Baylor College of Medicine Joint Center for Medical Genetics, The Chinese University of Hong Kong, Hong Kong, China; 5The Fertility Preservation Research Center, Department of Obstetrics and Gynaecology, The Chinese University of Hong Kong, Hong Kong, China

**Keywords:** paternity test, genome sequencing, prenatal testing, low-pass genome sequencing, single-nucleotide variants

## Abstract

Submission of a non-biological parent together with a proband for genetic diagnosis would cause a misattributed parentage (MP), possibly leading to misinterpretation of the pathogenicity of genomic variants. Therefore, a rapid and cost-effective paternity/maternity test is warranted before genetic testing. Although low-pass genome sequencing (GS) has been widely used for the clinical diagnosis of germline structural variants, it is limited in paternity/maternity tests due to the inadequate read coverage for genotyping. Herein, we developed rapid paternity/maternity testing based on low-pass GS with trio-based and duo-based analytical modes provided. The optimal read-depth was determined as 1-fold per case regardless of sequencing read lengths, modes, and library construction methods by using 10 trios with confirmed genetic relationships. In addition, low-pass GS with different library construction methods and 1-fold read-depths were performed for 120 prenatal trios prospectively collected, and 1 trio was identified as non-maternity, providing a rate of MP of 0.83% (1/120). All results were further confirmed via quantitative florescent PCR. Overall, we developed a rapid, cost-effective, and sequencing platform-neutral paternity/maternity test based on low-pass GS and demonstrated the feasibility of its clinical use in confirming the parentage for genetic diagnosis.

## 1. Introduction

Nowadays, with the rapid advancements in sequencing technologies, prenatal testing via genome sequencing (GS) methods has been widely introduced in clinical settings [1] to provide a higher diagnostic yield of genetic abnormalities compared to traditional tests among high-risk pregnancies [2,3,4]. In particular, trio-based (proband and biological parents) GS testing determines the mode of inheritance of genomic variants, assisting variant classification and the interpretation of clinical significance [4]. However, the submission of one or both non-biological parents would cause a misattributed parentage (MP), possibly resulting in misdiagnosis. A family-based exome sequencing study for genetic diagnosis identified 0.58% of MP, including non-paternity and non-maternity [5]. Based on the estimation from the American Society of Human Genetics, misattributed paternity occurs at a rate between 1 and 10% [6]. It is understood that the rate of MP might be increased with the increasing rate of adoption or gamete donation due to infertility. In prenatal diagnosis, the presence of MP might also prevent timely fetal disease diagnosis, leading to the difficulty of further management and decision-making.

Currently, a polymerase chain reaction (PCR)-based method utilizing short tandem repeats (STRs) serves as the gold-standard method for paternity testing [7]. However, challenges remain. For instance, stutter artifacts generated during amplification due to repetitive motifs and mutations in STRs could interfere with the probability of paternity calculation. In comparison, although single nucleotide polymorphism (SNP) typing has been recently adopted for forensic science by genotyping of a list/panel of SNPs [8], allele frequencies among different races have not been evaluated with the existing panels [9,10]. Recently, a microhaplotype with at least 2 SNPs within 200 bp has been introduced; however, it also relies on genotyping approaches, such as high read-depth sequencing (GS/ES) [11].

Low-pass GS characterized by low-coverage and high-throughput genome sequencing (0.1~4-fold read-depth) has demonstrated its capability and feasibility in the detection of copy number variants [12,13,14,15], structural rearrangements [16,17,18], and regions with the absence of heterozygosity [14,19]. It has been recently recognized as an application of germline structural variant detection by the American College of Medical Genetics and Genomics [20]. It has become a widely used, cost-effective test in clinical laboratories. However, unlike targeting sequencing of panels with pre-selected markers, the detection capability of targeted single-nucleotide variants (SNVs) via low-pass GS is limited. Low-pass GS relies on a shotgun or random sequencing of nearly the entire genome, which generates relatively even coverage across the genome. The variation in sequencing coverage between samples and batches can make it challenging to obtain adequate reads for determining genotypes at pre-determined sites (Figure 1A). In addition, genotyping using low-pass GS data is error-prone due to the lack of coverage. Heterozygous SNVs could be mistakenly detected by low-pass GS as homozygous SNVs due to insufficient reads supporting the alternate allele. Similarly, SNVs could be missed if the coverage of the mutant allele is insufficient (Figure 1B). Lastly, pre-determining regions with high coverage across different samples and batches could be problematic because they could represent biases caused by systematic errors during alignment.

Herein, we developed a rapid, cost-effective, and sequencing platform-neutral paternity test based on low-pass GS (one-fold read-depth). In addition, we validated its feasibility and robustness in prenatal and postnatal cases in duo analysis mode (designed for the submission of a pair of samples: proband and a presumed parent) and in trio analysis mode (for the submission of three samples: proband and two presumed parents), respectively.

## 2. Materials and Methods

### 2.1. Case Recruitment

The study protocol was approved by the Ethics Committee of the Joint Chinese University of Hong Kong–New Territories East Cluster Clinical Research Ethics Committee (CREC Ref. No. 2016.713 and 2021.218). Informed written consent for sample storage and genetic analyses was obtained from each participant. In this study, there were 130 trios recruited (with the presumed parents) for clinical genetic diagnosis in our laboratory (referred to as clinical prenatal trios), including 10 trios with confirmed biological relationship for method development (as Phase I) and 120 trios prospectively for method validation (as Phase II). The sample sources included products of conception, chorionic villi, or amniotic fluids.

### 2.2. DNA Preparation for Low-Pass GS

Genomic DNA from proband or the parents was extracted with DNeasy Blood and Tissue Kit (cat. number/ID: 69506, Qiagen, Hilden, Germany) and treated with RNase (Qiagen, Hilden, Germany). DNA was subsequently quantified with the Qubit dsDNA HS Assay Kit (Invitrogen, Carlsbad, CA, USA), and the DNA integrity was assessed via gel electrophoresis. All samples passing QC (>500 ng; OD260/OD280 > 1.8; OD260/OD230 > 1.5) were subsequently prepared for library construction in low-pass GS.

### 2.3. Low-Pass GS

We selected 10 trios with confirmed biological relationships for low-pass GS. Five trios (15 samples) were subjected to small-insert size library construction [3], and the other five were subjected to mate-pair library construction [21]. For small-insert size libraries, genomic DNA from each sample was sheared with the Covaris E220 Evolution Focused-Ultrasonicator (Covaris, Inc., Woburn, MA, USA) into sizes of 300~500 bp and then subjected to library construction using the MGIEasy FS DNA Library Prep kit according to the manufacturer’s protocol. Each library (per sample) was sequenced with paired-end 150 bp for a read-depth of ~4-fold on an MGISEQ-2000 platform (MGI Tech Co., Ltd., Shenzhen, China). For mate-pair library construction, 1 μg of genomic DNA from each sample was sheared (3~8 kb) with a HydroShear device (Digilab, Inc., Hopkinton, MA, USA) and subjected to library construction following our reported methods [21]. A minimum of 60 million read-pairs (100 bp in length; equivalent to 4-fold read-depth) for each case [22,23] were obtained on an MGISEQ-2000 platform (MGI).

In addition, to further evaluate the accuracy of using the optimal parameters for paternity detection, another 120 clinical trios were sequenced on MGISEQ-2000 platform (MGI), including 100 trios sequenced with small-insert libraries and 20 trios sequenced with mate-pair libraries for a minimum of 1-fold at paired-end 100 bp (Figure 2).

### 2.4. Determination of Paternity and Maternity

After data QC assessment, the read/read-pairs were aligned to the human reference genome (GRCh37) with Burrows–Wheeler Aligner (BWA) [24] with mem module. With SAMtools [25], the alignment file was then sorted by the aligned chromosomes and locations, and the reads that were likely generated from PCR duplication were removed. It was then reformatted by the Mpileup module from SAMtools to calculate the coverage and to determine the genotype of each genomic location. Loci with read(s) supporting a mutant base type were selected for further analysis. An SNV was defined if there were 5 to 20 reads covering that locus and over 2 reads supporting a mutant base type [19]. The genotype of this SNV was defined as homozygous if 100% of reads were supporting the mutant base type, whereas a heterozygous SNV was defined as 25 to 75% of reads supporting the mutant base type. Two modes of analysis were provided.

Two analytical models were presented: a duo mode and a trio mode. For the trio-based analysis mode, loci in which both parents were homozygous for different genotypes were selected (for instance, a locus where the father was with homozygous A, whereas the mother was with homozygous T). In theory, the proband should carry a heterozygous AT genotype. However, in low-pass GS setting, proband could also show a homozygous genotype similar to one of parents (Appendix A). It might be due to (a) one of the parents having a heterozygous genotype but mistakenly assigned as homozygous; (b) proband was detected as heterozygous but mistakenly assigned as homozygous; or (c) the genotype in one of the parents resulted from systematic error(s). Lastly, the proband may carry a heterozygous genotype (e.g., AG) but one base type (i.e., G) was from neither parent. In addition to these false SNV calling events, the main reason for the inconsistency of base-type inheritance between the proband and the presumed parent(s) was non-paternity and/or non-maternity. Therefore, we hypothesized that for paternity test (or maternity test), the inconsistent rate of base-type inheritance in non-paternity (non-maternity) would be significantly higher than that in a biological family. Among loci where both parents were homozygous for different genotypes, we further calculated the percentage of SNVs that were homozygous in the proband but with different genotypes from the presumed father (or mother), which served as the inconsistent rate of base-type inheritance.

In comparison, for the duo-based analytical mode, we hypothesized that in a locus, if it was homozygous in the presumed father/mother, in the proband, it was heterozygous with one allele identical with that of the parent or homozygous and was the same as the submitted parent. However, in low-pass GS setting, it might be homozygous in the proband, but the genotype was different from the parent potentially due to (a) it was heterozygous in that parent but mistakenly assigned as homozygous; (b) it was heterozygous in the proband but mistakenly assigned as homozygous; or (c) the genotype in one of them resulted from systematic error(s). In addition to these false SNV calling events, the main reason for the inconsistent base-type inheritance between the proband and the presumed parent was non-paternity and/or non-maternity. Therefore, we first selected the loci in which both proband and the presumed parent were homozygous, and among them, we calculated the percentage of SNVs that were with different genotypes between the proband and the presumed father to serve as the inconsistent rate of base-type inheritance (See Appendix A).

### 2.5. Data Simulation

To determine the precise cutoff for the paternity test, parental data from different families were randomized to form non-paternity (or non-maternity) families among the 10 trios. Sequencing data of different read-depths (such as 0.5-, 1-, 2-, 3-, and 4-fold) from 10 trios were used for evaluation. In addition, to determine the optimal sequencing parameters for paternity testing [such as read length, library construction, and sequencing mode (paired-end or single-end)], we used read1 from the paired-end sequencing data as single-end sequencing data, while 150 bp reads were trimmed into 100 bp to serve as sequencing data with shorter read-lengths.

In addition, to evaluate the performance of datasets sequenced with Illumina platform, 50 trios sequenced in NovaSeq 6000 System (Illumina, San Diego, CA, USA) with small-insert libraries were also randomly selected from the 1000 Genomes Project (1 KGP) [26] (Appendix A). The GS data in CRAM format were downloaded from the 1 KGP and converted into FastQ format, trimmed (paired-end 100 bp), and down-sampled to a minimum of 1-fold per sample.

### 2.6. Verification of Parental Inheritance

For the clinical samples in Phase I (10 trios) and Phase II (120 trios), parental inheritance was confirmed via quantitative fluorescence polymerase chain reaction (QF-PCR) with 100 ng DNA from each sample by using short tandem repeat (STR) markers located on chromosomes 13, 18, 21, X, and Y (Appendix A) [15]. For the 50 trios downloaded from the 1000 Genomes Project, we utilized the genotypes from each family member identified via high read-depth GS for confirmation (Appendix A) among SNPs commonly used for paternity tests [8].

## 3. Results

### 3.1. Establishment of Optimal Parameters for Prenatal Testing

To determine the optimal parameters for the paternity test, we first selected 10 prenatal trios with confirmed paternity and maternity and subjected them to low-pass GS with 2 types of library constructions. In addition, data simulation was performed for each sample to generate different sets of low-pass GS data with consistent sequencing parameters (i.e., read-lengths and sequencing modes) among the family members and with sequencing data of different read-depths (such as one-fold). In addition, we randomly assigned the paternal/maternal samples for each family to form a non-paternity and/or non-maternity family. Trio-based and duo-based modes were performed for each family with the same analytical parameters to calculate the inconsistent rates of paternal/maternal inheritance for comparison (Figure 3 and Appendix A, Appendix A).

Of note, at the read-depth of 0.5-fold, a large deviation in the inconsistent rates of inheritance among different duos and a much smaller difference in the inconsistent rates between biological duos and non-biological duos than those from higher read-depths occurred were observed. However, at the read-depth of 0.5-fold, most of the SNVs detected were likely contributed by systematic errors due to the insufficient read-depth; thus, it reduced the accuracy of the paternity test by using this method. The result indicated the feasibility of paternity/maternity testing with a one-fold read-depth for both trio-based and duo-based analysis regardless of read lengths (100 or 150 bp), sequencing modes (single-end or paired-end), and library construction methods (small-insert or mate-pair). For trio-based analysis with the settings of 1-fold, paired-end sequencing at 100 bp, and small-insert libraries, the average inconsistent rates of paternal inheritance among the 5 biological and 5 non-biological trios were 18.8% [standard deviation (SD): 1.89%] and 38.5% (SD: 1.19%), respectively, while the average inconsistent rates of maternal inheritance were 18.0% (SD: 3.03%) and 37.8% (SD: 1.12%), respectively. In comparison, for duo-based mode with the same setting, the average inconsistent rates of paternal inheritance among the 5 biological and 5 non-biological trios were 18.5% (SD: 0.67%) and 38.4% (SD: 1.02%), respectively, while the average inconsistent rates of maternal inheritance were 18.3% (SD: 0.46%) and 37.9% (SD: 1.00%), respectively. The inconsistent rates of paternal/maternal inheritance between the two analytical modes were consistent. In comparison, with the settings of 1-fold, paired-end sequencing at 100 bp, and mate-pair libraries, the results were highly consistent with the ones observed in the data from small-insert libraries (Figure 3). In order to be a platform-neutral test, we calculated the average and SD of the inconsistent rate of base-type inheritance for both paternal and maternal tests among all 10 trios (*n* = 20) for each mode. The average and SD for trio-based analysis were 18.1% and 2.1%, while the average and SD for duo-based analysis were 17.1% and 1.7%, respectively. Therefore, we set the cutoff of reporting a biological father/mother would be 24.5% (Z > 3) for trio-based analysis. For duo-based analysis, as the average values of the inconsistent rate of base-type inheritance between the biological father/mother and non-biological father/mother were much larger than the one from trio-based analysis (Figure 3), we set 25.6% (Z > 5) as the cutoff to report a biological father/mother in order to strengthen the analysis.

To determine the turnaround time (TAT) when the data were within the optimal setting (1-fold and paired-end 100 bp), the TAT required for each step was recorded. The total time required for the whole analysis was less than 1 h (Appendix A) for either mode of the analysis (trio-based or duo-based).

### 3.2. Validation among 120 Clinical Trios and 50 Trios from 1 KGP

To further validate the platform-neutral performance of low-pass GS in paternity/maternity tests among different methods of library constructions and different sequencing platforms, we subjected 100 clinical prenatal trios for sequencing with small-insert libraries from MGISeq-2000 and 20 prenatal clinical trios for sequencing with mate-pair libraries also from MGISeq-2000. In addition, we also randomly selected 50 trios sequenced with small-insert libraries from NovaSeq from 1 KGP by down-sampling the sequencing data to read a depth of 1-fold.

Paternity and maternity testing were performed in both trio and duo modes, respectively. Interestingly, all trios were reported as biological families except for case 22C1246. For 22C1246, the inconsistent rates of maternal inheritance by the trio-based and duo-based analysis were 38.1% and 37.7%, respectively, indicating the mother was not the biological mother. All clinical trios (*n* = 120) were subjected to QF-PCR for paternity/maternity validation, while among the 50 trios from 1 KGP, genotype information of the common SNPs among the proband and the presumed parents were used for confirmation (Appendix A). For case 22C1246, the STR marker confirmed that the mother was not the biological mother (Figure 4). A follow-up study indicated that the pregnancy was achieved via oocyte donation. The confirmation assays yielded a 100% consistent result with our testing method (Figure 4A,B and Appendix A).

Therefore, among the 120 clinical prenatal trios, the MP rate was 1 in 120 (0.83%).

### 3.3. Establishment of In-House Datasets of Recurrent SNVs Likely Resulted from Systematic Errors

As GS likely provides randomly distributed reads among the genome, the recurrent SNVs likely resulted from systematic errors generated during alignment. We would like to investigate the presence of such recurrent SNVs with the optimal read-depth of one-fold.

Among all 180 families (including 10 from Phase I and 170 from Phase II), for trio-based analysis, the average number of loci that were homozygous in both parents but with different genotypes and with 5 to 20 reads supporting in the proband was ~707 for trio-based analysis. Among them, an average of 126 SNVs were regarded as inconsistent with paternal/maternal inheritance in both paternity and maternity testing. Overall, 663 loci were detected more than once among these 360 tests (paternity and maternity). The average number of detecting recurrent SNVs per analysis was ~5 (<1%, 5/707) for paternity and ~4 (<1%, 4/707) for maternity, respectively.

For duo-based analysis, the average number of detected SNVs that were homozygous in the proband and presumed father/mother was ~11,158. In addition, an average of 2097 SNVs were regarded as inconsistent with parental inheritance per analysis. A total of 15,325 and 14,555 loci were detected more than once in proband-father and proband-mother analysis, respectively (Appendix A). The percentage of these recurrent SNVs per test was ~2.1% (229/11,158) in the proband-father analysis and ~2.0% (218/11,158) in the proband-mother analysis.

Although the percentage of such “recurrent SNVs” were relatively low for either analytical mode, to further improve the accuracy rate, we established a dataset of all recurrent SNVs identified from either mode. It served as an in-house dataset to filter out the SNVs that likely resulted from systematic errors.

## 4. Discussion

In this study, we developed robust paternity testing based on low-pass GS data and validated the performance from prenatal and postnatal cases with different sample sources. It is a rapid (an overall TAT of data analysis at <1 h), sequencing platform-neutral (regardless of sequencing parameters), and cost-effective (with read-depth of as low as 1-fold) paternity/maternity test.

Prenatal genetic testing via low-pass GS has been widely performed for germline structural variants detection [20]; however, it is limited in genotyping due to insufficient coverage leading to the difficulty of paternity/maternity testing. Unlike STR-based and SNP-based technologies, the accuracy of which is highly dependent on the selection and amplification of specific genetic markers [10,27], we performed the analysis in trio-based or duo-based genome-wide mode. In addition, to minimize the effect of false positive or false negative detection of SNVs, we established a baseline of inconsistent rates of paternal/maternal inheritance by using 10 trios with confirmed biological relationships and investigated the spectrum of inconsistent rates of paternal/maternal inheritance with non-paternity/maternity families by randomly assigning the parents to the probands. The robust performance was further confirmed by using 170 trios sequenced with different library constructions and sequencing platforms. To evaluate the effect contributed by systematic errors (such as alignment), we identified 593 recurrent loci via trio-based analysis among all analyzed trios. There were only ~1% of the overall available loci per test. In comparison, via duo-based analysis, due to the filter criteria of SNV detection only requiring 2 samples, nearly 10 times the loci were available for the analysis. However, the percentage of detecting recurrent SNVs was only ~2% for paternity/maternity testing. Taken together, our results not only echoed that GS provided a randomly distributed coverage among the genome but also demonstrated that the effect contributed by systematic errors was minimal. Nonetheless, we established a database to include these recurrent loci, and for further application, the loci curated in this dataset would be filtered out.

Two modes were provided in this testing, trio-based and duo-based, which were based on different hypotheses of variant inheritance. For each mode, the TAT of data analysis was less than 1 h. Although only one mode might be sufficient to indicate the paternity/maternity for each family, integration of two pipelines is also suggested when there is a trio submitted in order to double confirm the results. In particular, two pipelines shared most of the analytical steps (such as alignment and reformatted); thus, the TAT of integration or running the pipelines in parallel would be also less than 1 h.

It is noteworthy that families with children without genetic connections are more and more widespread due to the increasing rates of births that recur to gamete donation and surrogacy, together with adoptions [28]. According to ESHRE registries, more than 178,027 oocyte donation cycles have been performed only in Europe by 2011, and the number has steadily increased [29]. In this study, we identified one of the 120 clinical trios prospectively collected was with non-maternity, providing a rate of MP as 0.83%, similar to a family-based exome sequencing study [5] (MP = 0.58%). Therefore, quick and accurate paternity/maternity testing as a QC test to confirm parentage for genetic diagnosis and to avoid sample mix-up is needed.

In this study, all results have been confirmed by a gold-standard method QF-PCR for the 130 clinical trios and for the genotype comparison for trios from the 1000 Genomes Project. This indicates that this test is also able to provide confirmation if the family only looks for a paternity/maternity test, although validation with a larger scale sample size would be warranted. In particular, in this study, the feasibility of applying our method as a clinical test has been demonstrated through validation with different clinical sample sources, including products of conception and prenatal and postnatal samples.

Overall, we developed a rapid, cost-effective, and sequencing platform-neutral paternity/maternity test based on low-pass GS (as low as 1-fold read-depth) with two analytical modes provided (trio-based and duo-based), and we demonstrated its robust performance with data sequenced from different library construction methods and platforms with further confirmation via QF-PCR.

## Figures and Tables

**Figure 1 genes-14-01357-f001:**
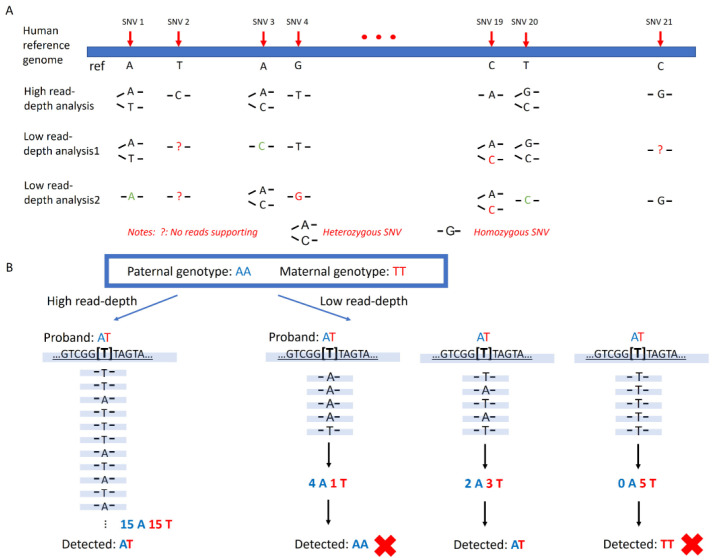
Illustration of the difficulty of SNV detection via low-pass genome sequencing. (**A**) Comparison of detecting pre-selected sites via high read-depth GS and low-pass GS. For the analysis with high read-depth GS, genotyping is precise to indicate heterozygous or homozygous SNVs due to the adequate read-depth. Whilst by low-pass GS, due to the randomly distributed reads, SNVs might not be detected (indicated by question mark), misassigned as homozygous (indicated by green font), or incorrectly detected (indicated by red font) between different batches. (**B**) Possible scenarios of homozygous SNV detection caused by insufficient reads with low-pass GS. For the locus, the genotype in the proband should be heterozygous AT based on the genotypes in the parents. Apparently, an equal number of reads supporting the reference and mutant base types are shown in high read-depth GS. In the scenarios of low-pass GS, the genotype might be misassigned as homozygous AA or TT due to the insufficient reads in supporting the reference (in blue fonts) or mutant base type (in red fonts).

**Figure 2 genes-14-01357-f002:**
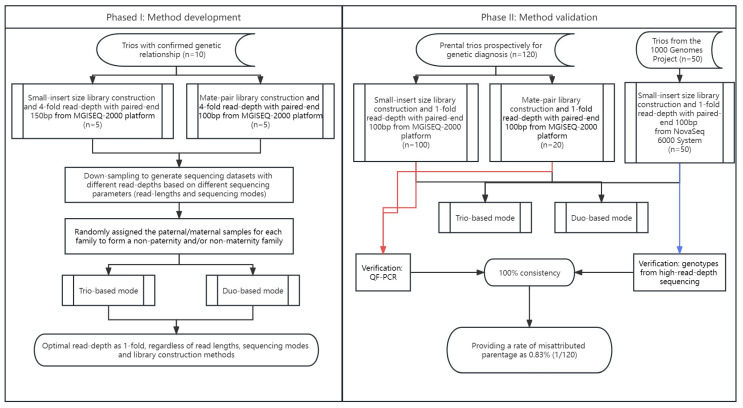
Workflow of method development and evaluation with low-pass genome sequencing for the paternity test. Phase I (method development) and Phase II (method validation) are shown in the left and right panels, respectively. In Phase II, the 120 prospectively collected trios were subjected to validation of the paternity/maternity via QF-PCR (indicated by red line), while the 50 trios downloaded from the 1000 Genomes Project were subjected for paternity/maternity confirmation via genotyping analysis (indicated by blue line).

**Figure 3 genes-14-01357-f003:**
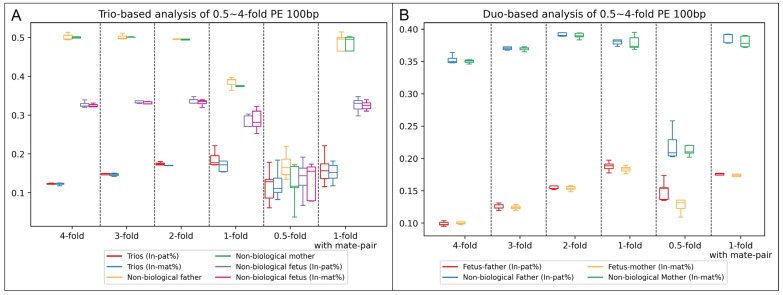
Determination of the optimal read-depth required for the analysis. Boxplot of the inconsistent rate of parental inheritance among paired-end 100 bp data with different read-depths and with small insert library and mate-pair library constructions in trio-based analysis (**A**) and in duo-based analysis (**B**). (**A**) In trio-based analysis, the inconsistent rates of paternal and maternal inheritance in biological trios are indicated in red and blue boxes, respectively. In addition, the inconsistent rates of paternal inheritance (in trios with non-biological father) and maternal inheritance (in trios with non-biological mother) are shown in yellow and green boxes, respectively. Furthermore, the inconsistent rates of paternal and maternal inheritance in trios with non-biological fetuses are indicated in purple and pink boxes, respectively. (**B**) In duo-based analysis, the inconsistent rates of paternal inheritance in fetus–father duos and fetus–non-biological-father duos are indicated in red and blue boxes, respectively, whereas the inconsistent rates of maternal inheritance in fetus–mother duos and fetus–non-biological-mother duos are indicated in yellow and green boxes, respectively.

**Figure 4 genes-14-01357-f004:**
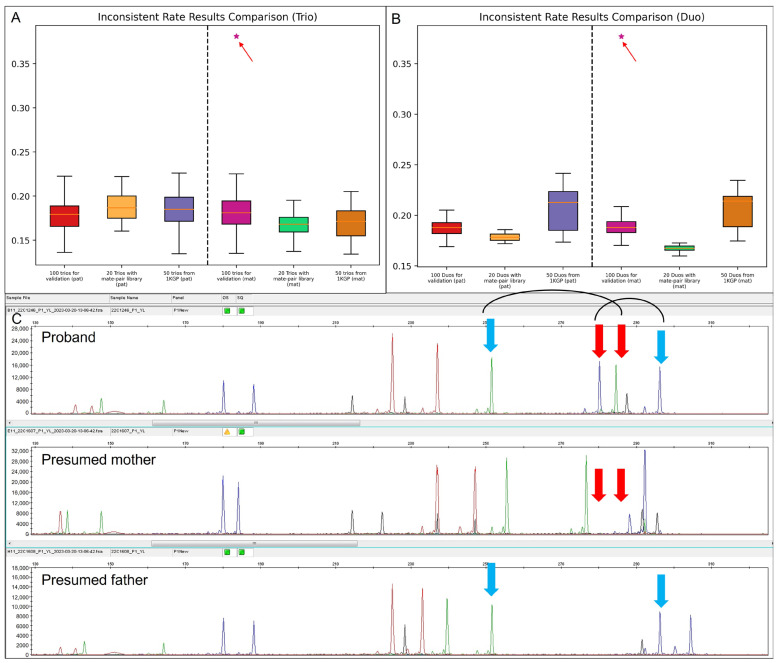
Evaluation of the performance among 170 trios with data from different library construction methods and sequencing platforms. Boxplot of the inconsistent rate of parental inheritance among 100 trios with small-insert libraries and sequenced by MGISEQ-2000, 20 trios with mate-pair libraries and sequenced by MGISEQ-2000, and 50 trios with small-insert libraries and sequenced by NovaSeq 6000 System in trio-based analysis (**A**) and in duo-based analysis (**B**). The outlier (shown as *) in both analyses is indicated by the red arrow in each figure. (**C**) QF-PCR with STR marker for the validation in family of 22C1246 (proband), 22C1607 (the presumed mother), and 22C1608 (the presumed father). Two pairs of loci are shown to indicate non-maternity. Each pair of a locus is linked; the allele in the proband inherited from the father is indicated by a blue arrow, while the other allele in the proband is indicated by a red arrow, which is not presented in the mother.

## Data Availability

The source code is available at https://github.com/ChloeL2023/LpPat.git (accessed on 25 June 2023).

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
