# Peer review of "Low-Pass Genome Sequencing-Based Detection of Paternity: Validation in Clinical Cytogenetics"

_genes, 2023, doi:10.3390/genes14071357_

Round 1

Reviewer 1 Report

Altogether, the authors performed an interesting work, validating a paternity/maternity test based on low-pass GS on prenatal and postnatal duo and trios.

However, in my opinion, some points should be clarified:

       Introduction:

1)      “thus, it shows the potential to be used for paternity testing”: I suggest removing this sentence since the high error rate in the identification of single nucleotide variant/small indels

Material and Methods:

1)      A diagram in the main text showing the complete methodology workflow to develop and validate the test, including the cases used, is needed to understand the work.

2)      Specify what is meant by “clinical trios”

3)      Phase I and Phase II: to be introduced earlier in the text or delete

4)      Results, Figure 2:

-          explain the different colors of the box plot in the legend

-          why does the inconsistency rate seem lower at 0.5X than at higher read depths?

-          Duo-based analysis: is there an error in the colors and legend of the 0.5X and 1X part?  

5)      Discussion:

-          It is noteworthy that families have children: substitute “have” with “with”

-          In this study, all results have been confirmed: add how these results were confirmed

-          particularly for those 130 clinical trios that were confirmed by QF-PCR: delete “that were”

-          able to provide a confirmation: delete “a”

English level is appropriate

Author Response

Response to the Reviewer 1:

Point 1: Altogether, the authors performed an interesting work, validating a paternity/maternity test based on low-pass GS on prenatal and postnatal duo and trios.

Response 1: We sincerely appreciate the suggestions and comments from the Reviewer, and we would like to address your questions point to point as below.

Point 2: However, in my opinion, some points should be clarified:

       Introduction:

  • “thus, it shows the potential to be used for paternity testing”: I suggest removing this sentence since the high error rate in the identification of single nucleotide variant/small indels

Response 2: We agree with the Reviewer, and we removed this sentence in the revised manuscript. Please refer to Line 6 of the third paragraph in Page 2.

Point 3: Material and Methods: A diagram in the main text showing the complete methodology workflow to develop and validate the test, including the cases used, is needed to understand the work.

Response 3: Thanks for the suggestion. We included a diagram showing the workflow of this study including method development (Phase I) and method validation (Phase II). Please refer to the new Figure 2 shown in Page 5 of the resubmitted manuscript.

Point 4: Specify what is meant by “clinical trios”

Response 4: We regret for the misleading. They are trios recruited from clinically genetic diagnosis in our laboratory (referred as clinical prenatal trios). We refined the sentence as “In this study, there were 130 trios recruited (with the presumed parents) during clinically genetic diagnosis in our laboratory (referred as clinical prenatal trios)…”, in Page 4 under the subtitle of 2.1 case recruitment.

Point 5: Phase I and Phase II: to be introduced earlier in the text or delete

Response: We agree with the Reviewer. We included this information in Page 4 under the subtitle of 2.1 case recruitment as “including 10 trios with confirmed biological relationship for method development (as Phase I) and 120 trios prospectively for method validation (as Phase II).”

Point 6: Results, Figure 2: explain the different colors of the box plot in the legend

Response 6: We provided additional figure legend as “(A) In trio-based analysis, the inconsistent rates of paternal and maternal inheritance in biological trios are indicated in red and blue boxes, respectively. In addition, the inconsistent rates of paternal inheritance (in trios with non-biological father), and maternal inheritance (in trios with non-biological mother), are shown in yellow and green boxes, respectively. Furthermore, the inconsistent rates of paternal and maternal inheritance in trios with non-biological fetus are indicated in purple and pink boxes, respectively. (B) In duo-based analysis, the inconsistent rates of paternal inheritance in fetus-father duos and fetus-non-biological-father duos are indicated in red and blue boxes, respectively, whereas the inconsistent rates of maternal inheritance in fetus-mother duos and fetus-non-biological-mother duos are indicated in yellow and green boxes, respectively.” Please refer to the updated figure legend of the Figure 3 (Figure 2 in the original version) in Page 7.

Point 7: Figure 2, why does the inconsistency rate seem lower at 0.5X than at higher read depths?

Response 7: Thanks for pointing out this. At the read-depth of 0.5-fold, most of the SNVs detected might be contributed by systematic errors due to the insufficient read-depth. It led to a large deviation of the inconsistent rates of inheritance among different duos, and a much smaller difference of the inconsistent rates between biological-duos and non-biological duos than that from higher read-depths. Thus, it reduced the accuracy of paternity test using this method. We further included this information as “Of note, at the read-depth of 0.5-fold, a large deviation of the inconsistent rates of in-heritance among different duos, and a much smaller difference of the inconsistent rates between biological-duos and non-biological duos than that from higher read-depths. However, at the read-depth of 0.5-fold, most of the SNVs detected were likely contributed by systematic errors due to the insufficient read-depth; thus, it reduced the accuracy of paternity test by using this method.”. Please refer to Page 7.

Point 8: Figure 2, Duo-based analysis: is there an error in the colors and legend of the 0.5X and 1X part?  

Response 8: We regret for the transcription error. We updated the Figure with corrected legend (Figure 3 in Page 7).

Point 9: Discussion:

-          It is noteworthy that families have children: substitute “have” with “with”

-          In this study, all results have been confirmed: add how these results were confirmed

-          particularly for those 130 clinical trios that were confirmed by QF-PCR: delete “that were”

-          able to provide a confirmation: delete “a”

Response 9: Appreciate the comments and suggestions. We revised them accordingly.

Reviewer 2 Report

This is an interesting paper, well written, and significant content.

I reviewed the manuscript and found it well written, the only issue is that it needs to be summarized otherwise no other corrections are needed

Author Response

Point 1: This is an interesting paper, well written, and significant content.

I reviewed the manuscript and found it well written, the only issue is that it needs to be summarized otherwise no other corrections are needed.

Response 1: We sincerely appreciate the comments from the Reviewer. We also agree with the Reviewer a summary of the workflow would be important, and in the resubmitted version, we have included a diagram showing the workflow of this study including method development (Phase I) and method validation (Phase II). Please refer to the new Figure 2 shown in Page 5 of the resubmitted manuscript.